# Modulation of Neurexins Alternative Splicing by Cannabinoid Receptors 1 (CB1) Signaling

**DOI:** 10.3390/cells14130972

**Published:** 2025-06-25

**Authors:** Elisa Innocenzi, Giuseppe Sciamanna, Alice Zucchi, Vanessa Medici, Eleonora Cesari, Donatella Farini, David J. Elliott, Claudio Sette, Paola Grimaldi

**Affiliations:** 1Department of Biomedicine and Prevention, University of Rome Tor Vergata, 00133 Rome, Italy; elisa.inno92@gmail.com (E.I.); azucchi97@gmail.com (A.Z.); donatella.farini@uniroma2.it (D.F.); 2Departmental Faculty of Medicine, UniCamillus—Saint Camillus International University of Health and Medical Sciences, 00131 Rome, Italy; giuseppe.sciamanna@unicamillus.org; 3Istituto di Ricovero e Cura a Carattere Scientifico (IRCCS), San Camillo Hospital, 30126 Venice, Italy; 4Department of Neuroscience, Section of Human Anatomy, Catholic University of the Sacred Heart, 00168 Rome, Italy; vanessamedici3@gmail.com (V.M.); claudio.sette@unicatt.it (C.S.); 5GSTeP Organoids Research Core Facility, Fondazione Policlinico A. Gemelli Istituto di Ricovero e Cura a Carattere Scientifico (IRCCS), 00168 Rome, Italy; eleonora.cesari@gmail.com; 6Newcastle University Centre for Cancer, Newcastle University Institute of Biosciences, Newcastle NE1 3BZ, UK; david.elliott@ncl.ac.uk

**Keywords:** endocannabinoid system, neurexin, alternative splicing, cannabinoid receptors, hippocampus

## Abstract

Synaptic plasticity is the key mechanism underlying learning and memory. Neurexins are pre-synaptic molecules that play a pivotal role in synaptic plasticity, interacting with many different post-synaptic molecules in the formation of neural circuits. Neurexins are alternatively spliced at different splice sites, yielding thousands of isoforms with different properties of interaction with post-synaptic molecules for a quick adaptation to internal and external inputs. The endocannabinoid system also plays a central role in synaptic plasticity, regulating key retrograde signaling at both excitatory and inhibitory synapses. This study aims at elucidating the crosstalk between alternative splicing of neurexin and the endocannabinoid system in the hippocampus. By employing an ex vivo hippocampal system, we found that pharmacological activation of cannabinoid receptor 1 (CB1) with the specific agonist ACEA led to reduced neurotransmission, associated with increased expression of the Nrxn1–3 spliced isoforms excluding the exon at splice site 4 (SS4−). In contrast, treatment with the CB1 antagonist AM251 increased glutamatergic activity and promoted the expression of the Nrxn variants including the exon (SS4+) Knockout of the involved splicing factor SLM2 determined the suppression of the exon splicing at SS4 and the expression only of the SS4+ variants of Nrxns1–3 transcripts. Interestingly, in SLM2 ko hippocampus, modulation of neurotransmission by AM251 or ACEA was abolished. These findings suggest a direct crosstalk between CB1-dependent signaling, neurotransmission and expression of specific Nrxns splice variants in the hippocampus. We propose that the fine-tuned regulation of *Nrxn1*–*3* genes alternative splicing may play an important role in the feedback control of neurotransmission by the endocannabinoid system.

## 1. Introduction

Synaptic modifications of neural connections underlie the cognitive basis of learning and memory. Synapses are shaped by adhesion molecules that initiate the establishment of synapses and form their properties and plasticity. Neurexins (Nrxns) are cell-adhesion molecules, primarily expressed at the pre-synaptic site [1], which play important roles in signal transmission and information processing at synapses. Three *Nrxn* homologous genes (*Nrxn1*–*3*) have been identified in mammals, each of which gives rise to alpha- (α-Nrxn) and beta- (β-Nrxn) isoforms via independent promoters [2]. α-Nrxns and β-Nrxns encode type I transmembrane proteins with identical intracellular domains but different extracellular domains [3]. α-Nrxns contain six LNS (laminin-neurexin sex hormone binding globulin) domains with three interspersed epidermal growth factor-like (EGF) domains. The shorter β-Nrxns contain only one LNS domain, the αLNS6, and share an identical carboxyl terminus with α-Nrxns. Alternative splicing (AS) further increases the complexity of α- and β-Nrxn isoforms. AS regulates the functional activity of neurexins and plays a key role in determining the specificity of their interaction with various post-synaptic ligands. α-Nrxns can be alternatively spliced at six conserved splice sites (SS#1 to SS#6), while only SS#4 and SS#5 are present in β-Nrxn pre-mRNA sequences. This structural organization allows the generation of a plethora of splice isoforms [4,5,6]. In particular, the SS4 site is highly conserved in all *Nrxn* pre-mRNAs and is best characterized at the functional level. Inclusion/exclusion (+/− SS4) of a 90 bp exon cassette was shown to determine the interaction with specific post-synaptic ligands. Nrxns SS4− bind to dystroglycan, CIRL/latrophilin and LRRTMs [7,8,9,10], whereas Nrxns SS4+ bind to cerebellins [11,12]. Moreover, SS4− and SS4+ Nrxns exhibit distinct affinities for different isoforms of neuroligins, creating a “splice code” for synaptic interactions [13,14,15]. *Nrxn* SS4 splice variants are differentially expressed in various brain regions [16], exhibit a diurnal cycle [17] and are modulated by neuronal activity [18,19,20] and during cortical development [21].

This mechanism is particularly relevant in brain regions involved in cognitive processing, such as the hippocampus, where synaptic plasticity forms the cellular basis of learning and memory. Activity-dependent alternative splicing of synaptic genes, including Nrxns, has been shown to dynamically shape the postsynaptic receptor landscape and influence memory encoding and long-term information storage [22,23,24].

Endocannabinoids (ECs) also play a pivotal role in regulating synaptic functions [25]. They exert their biological function by activating cannabinoid receptors type 1 (CB1) and type 2 (CB2), and together with the enzymes required for ECs synthesis, membrane transport and hydrolysis, they form the endocannabinoid system [26,27,28,29]. CB1 is particularly abundant in the central nervous system (CNS), especially within regions associated with memory, emotion and motor control, such as the hippocampus, cortex and basal ganglia [30,31]. It is expressed on presynaptic neurons and mediates retrograde signaling of endocannabinoids, regulating neurotransmission and plasticity. It is well established that physiological activation of neurons induces the synthesis of ECs in postsynaptic neurons. In turn, ECs act as retrograde signaling agents activating pre-synaptic CB_1_ receptors and leading to suppression of neurotransmitter release in a highly selective and restricted manner [32]. The modulation of neurotransmitter release regulates the synaptic strength and contributes to synaptic plasticity. Accordingly, alterations in the ECS have been associated with different psychiatric disorders, such as depression, anxiety, autism and addiction [33,34,35]. Interestingly, crosstalk between Nrxns and retrograde signaling by the endocannabinoid system (ECS) has been highlighted [36]. Specifically, in the hippocampus, the SS4− splice variant of β-Nrxns was shown to modulate the synaptic strength of excitatory synapses by inhibiting the postsynaptic synthesis of the endocannabinoid 2-AG, thus promoting neurotransmission [36].

Herein, we aimed to better characterize the crosstalk between *Nrxns* and ECS signaling in the hippocampus by studying the potential relationship between CB1-dependent neurotransmission and Nrxns alternative splicing at SS4. By using a pharmacological approach in the ex vivo system, we showed that modulation of CB1 activity, besides inducing stimulation/inhibition of neurotransmission, was associated with the expression of specific Nrxns splice variants. Deletion of the involved splicing factor Slm2 abolished the SS4 alternative splicing of Nrxns, and concomitantly, it suppressed the CB1-dependent neural activity, suggesting that alternative splicing of Nrxns could be a mechanism by which endocannabinoids exert a fine-tuned regulation of neurotransmission in the hippocampus.

## 2. Materials and Methods

### 2.1. Animals

C57BL/6 wild-type (WT) and Slm2 ko mice were randomly assigned and housed in standard clear plastic cages, kept in light/dark cycle of 12:12 h and ventilation 10–20 times/h, with ad libitum water and food. Mice were kept in social groups at a constant temperature of 20 ± 2 °C and relative humidity of 50 ± 10%. All efforts were made to minimize the number of animals used and their suffering.

### 2.2. Primary Hippocampal Neuronal Cultures

Neuronal cultures were prepared as previously reported [37] from hippocampi at embryonic day 17.5 (E17.5) of C57BL/6 WT mice. The brain was removed, and hippocampus was freed of meninges, washed with PBS and centrifuged for 2 min at 1000 rpm. The tissue was resuspended and incubated for 5′ min at 37 °C with trypsin (Sigma-Aldrich T8003 0.25 mg/mL, Saint Louis, MO, USA), followed by the addition of trypsin inhibitor (Sigma-Aldrich T9003 0.52 mg/mL, Saint Louis, MO, USA) and DNase I (Sigma-Aldrich D5025 0.08 mg/mL, Saint Louis, MO, USA). Digested tissue was mechanically dissociated, centrifuged at 1000 rpm for 10 min and the dissociated cells were plated at a density of 1 × 10^6^ cells on poly-l-lysine (Sigma-Aldrich P2636 25 μg/mL, Saint Louis, MO, USA), coated 35 mm plates (Corning 430165, New York, NY, USA) in Neurobasal medium (Gibco, cat. no. 21103-049, Carlsbad, CA, USA) supplemented with 2% B27 (Gibco, cat. no. 17504-044, Carlsbad, CA, USA), 2 mM glutamine (Lonza BE17-605E, Walkersville, MD, USA) and 50 U/mL penicillin/streptomycin solution (Lonza DE17-602E, Walkersville, MD, USA). Cells grew at 37 °C and 5% CO_2_, changing half medium every 3 days from the seeding to the cell collection at T7, when we halved the concentration of B-27 supplement. Hippocampal neuronal cultures were treated with vehicle (DMSO) and drugs ACEA (1 µM) or AM251 (2 µM) for 5, 15, 30 min. These short time points of treatment were used for Western blot analysis of intracellular signaling, particularly Akt phosphorylation. Such signaling events are rapid and transient, typically occurring within minutes after receptor activation. Thus, the chosen time course allows monitoring of the onset and duration of CB1-triggered PI3K/Akt pathway activation in hippocampal neurons.

### 2.3. Drugs

Arachidonyl-2′-chloroethylamide (ACEA Tocris, 1319, 10 μM, Bristol, UK); AM251 (Tocris 1117, 10 μM, Bristol, UK) were dissolved in dimethyl sulfoxide (DMSO) at the concentration of 10 mM and diluted 1:10^3^/10^4^ in experimental conditions.

Picrotoxin (Tocris code n. 1128, 10 μM, Bristol, UK); CNQX (6-Cyano-7-nitroquinoxaline-2,3-dione disodium, Tocris, code n. 1045, 10 µM, Bristol, UK) and APV (2-APV,D-APV,D-2-amino-5-phosphonovalerate, Tocris code n. 0106, 10 µM, Bristol, UK).

### 2.4. Hippocampal Slices Stimulation and Recordings

Hippocampi were isolated from C57BL/6 wild-type (WT) and Slm2 ko one-month-old mice. Horizontal hippocampal slices (300 μm) were cut in ice-cold solution containing (in mM) 85 NaCl, 75 Sucrose, 2.5 KCl, 1.3 NaH_2_PO_4_, 24 NaHCO_3_, 0.5 CaCl_2_, 4 MgCl_2_, 25 D-Glucose saturated with 95% O_2_/5% CO_2_. Slices were stored for 30 min at 34 °C in ACSF bubbled with 95% O_2_ + 5% CO_2_ containing (in mM): NaCl (126), NaHCO_3_ (26), KCL (2.5), NaH_2_PO_4_ (1.25), CACl_2_ (2), MgSO_4_ (2), Glucose (10). Acute slices were transferred to a recording chamber continuously superfused with oxygenated ACSF (1.5 mL/min) maintained at 30.5 °C. Cells were visualized with a 40× water-immersion objective (LumpPlanFI, Olympus, Japan) and by an infrared camera EM-CCD camera (ImagEm, Hamamatsu, Japan). Patch-clamp recordings were made on hippocampal subiculum neurons by borosilicate glass pipette (3–5 MΩ) pulled with a micropipette puller (P97, Sutter Instruments, Novato, CA, USA) and filled with an internal solution consisting of the following (in mM): KMeSO_4_ (140), KCL (10), HEPES (10), Mg_2_ATP (2), Na_3_GTP (0.4); pH adjusted to 7.25 with KOH. Evoked excitatory postsynaptic currents (eEPSCs) were triggered at 0.1 Hz rate by electrical stimulation with a bipolar electrode positioned in the CA1 region. Electrophysiological recordings were acquired by a Multiclamp 700b amplifier, Digidata 1550A and pClamp 10.4 software (Molecular Devices, San Jose, CA, USA). Spontaneous and evoked EPSCs were recorded in pharmacological isolation by means of bath application of GABAa receptors antagonist Picrotoxin (50 µM, 10 min). Hippocampal slices were treated with vehicle (DMSO) and drugs ACEA and AM251 for 1 h as of recordings and 6 h for Nrxns molecular analyses. The 1 h treatment was applied before electrophysiological recordings (e.g., evoked and spontaneous EPSCs). This duration ensures sufficient time for drug diffusion and receptor interaction while still reflecting acute synaptic responses without inducing compensatory homeostatic changes that might occur with prolonged exposure. A shorter treatment time (1 h) was used to better preserve the neural vitality of the slices during the recording process. This duration was considered sufficient to trigger and observe the acute drug effect on synaptic transmission

The 6 h treatment was chosen for molecular assays such as RT-PCR, RT-qPCR and alternative splicing analysis of neurexins. Changes in gene expression and splicing patterns require longer timeframes to allow transcriptional and post-transcriptional mechanisms to occur. This time window is ideal for observing CB1-mediated modulation of alternative splicing via the SLM2 splicing factor.

For eEPSCs, recording events were triggered and collected at 0.1 Hz. At least 30 events were collected, and averages were collected for both control condition and drug application. For sEPSCs recordings, events were collected, analyzing at least 5 min recordings for both baseline and drug application. All data were collected only after cell stabilization was achieved. The experiment was initiated, and pharmacological procedures commenced only when the cell showed clear stabilization of parameters such as membrane potential and input resistance following whole-cell break-in (at least >5 min). Variation in the parameters exceeding 20% causes the experiments to be discarded.

### 2.5. RT-PCR Analysis and Quantitative Real-Time PCR (RT-qPCR)

Total RNA was extracted from hippocampal slices from young male mice of 1 month. Total RNA was extracted using Trizol reagent (Ambion life Technologies, Carlsbad, CA, USA) according to the manufacturer’s instructions, and 1 μg was used for retrotranscription (RT) using M-MLV reverse transcriptase (Ambion life Technologies, Carlsbad, CA, USA). Control reactions omitting M-MLV reverse transcriptase were also carried out. 20 ng of cDNA produced by the RT reaction was used as template for semiquantitative PCR analysis (RT-PCR) (GoTaq, Promega, Madison, WI, USA) or 12 ng for quantitative Real Time PCR analysis (RT-qPCR), using PowerUp SYBR Green Master Mix (Applied Biosystems, Carlsbad, CA, USA) and Applied Biosystems StepOnePlus Real-Time PCR system (Applied Biosystems) according to the manufacturer’s instructions. RT-PCR protocol was 95 °C 5′, 95 °C 30″, 60 °C 30″, 72 °C 30″, 72 °C 5′ repeated for 30 amplification cycles. RT-qPCR protocol: 95 °C 2′, 95 °C 15″, 60 °C 30″ repeated for 35 amplification cycles. RT-qPCR conditions are as in Ref. [37]. All the primers used are listed in Table 1. Nrxns splice variants were detected by RT-PCR using specific primers and analyzed on agarose gel. The +/−SS4 splicing variants were quantitated by densitometric analysis by calculating the (percentage of transcripts that include a specific exon (SS4+) of the total RNA (percentage spliced in, PSI). 

### 2.6. Western Blotting

For Western analysis, protein extracts were obtained from hippocampal slices or from cultured hippocampal neurons. Samples were lysed in 1% Triton X-100, 150 mM NaCl, 15 mM MgCl2, 15 mM EGTA, 10% Glycerol, 50 mM Hepes (pH 7.4) with protease inhibitors (Roche, 11873580001, Penzberg, BY, Germany). Proteins were separated by SDS–10% polyacrylamide gel electrophoresis and transferred to nitrocellulose membrane (Amersham, Little Chalfont, Buckinghamshire, UK). The membrane was blocked in PBS–5% skim milk powder for 1 h. Incubation of the membrane with the primary antibody was carried out at 4 °C o/n in PBS–5% BSA and then with the appropriate horseradish peroxidase-conjugated secondary antibody (Santa Cruz, Dallas, TX, USA). Anti-phospho Akt (Ser-473) (AB-10521) and anti-Tubulin (AB-J3066) were from Immunological Sciences, and anti-CB1 was from Cayman Chemical (101500). The horseradish peroxidase conjugate was detected by chemiluminescence (Bio-Rad, Hercules, CA, USA) with an ECL Kit (Amersham, Little Chalfont, Buckinghamshire, UK).

### 2.7. CB1 Immunohistochemistry

The whole brains of one-month-old mice were perfused with 4% PFA and paraffin embedded. Sections from hippocampus were deparaffinized, rehydrated and incubated overnight at 4 °C with primary antibody anti-CB1 diluted in DAKO ChemMateTM Antibody Diluent (1:100) (Dako, Glostrup, Denmark), extensively washed in PBS and then incubated for 1 h at room temperature with peroxidase-conjugated secondary antibodies and revealed by DAB substrate (Dako, Glostrup, Denmark). Sections were then counterstained with hematoxylin.

### 2.8. Statistical Analysis

Results were expressed as arithmetic mean ± standard deviation (SD). Student’s *t*-test was used to evaluate differences between two independent groups, whereas one-way ANOVA test was used to analyze differences among more than two independent groups, followed by Bonferroni test. All statistical tests were carried out using GraphPad Prism statistical analysis software package, version GraphPad Prism 6.0.

Densitometric analysis was performed using ImageJ-win64 or Image Lab 6.0.

The number of independent experiments was indicated in each figure legend. *p* values < 0.05 (*), <0.01 (**) and <0.001 (***) *p* < 0.0001 (****) were significant.

## 3. Results

### 3.1. Expression of Functional CB1 Receptors in Hippocampus

CB1 is the most abundant G protein-coupled receptor expressed in the brain, and its stimulation leads to modulation of neurotransmission. We examined the expression of CB1 receptor protein in the hippocampus by immunohistochemistry, using whole brains from one-month-old mice. Staining with antibody anti-CB1 of sections corresponding to the hippocampal region showed that the receptors were mainly expressed by pyramidal layer (PL) (Figure 1A).

Next, to investigate if treatment with agonist/antagonist of CB1 could modulate the expression level of the receptors, hippocampal slices from one-month-old mice were treated with ACEA or AM251 for 6 h. At the end of the treatment protein extracts were prepared and analyzed by Western blotting. Figure 1B shows that CB1 protein level did not change following drug treatment.

The synaptic transmission CB1-dependent was then investigated by electrophysiology, recording both evoked (eEPSCs) and spontaneous (sEPSCs) excitatory postsynaptic events by using CB1 agonist and antagonist. Glutamatergic fibers were electrically stimulated by a bipolar electrode placed at CA1 regions and recorded in subiculum neurons (as reported in Figure 2A) in the presence of picrotoxin (50 µM, 10 min) to avoid the effect of GABAergic tonic release. Bath application of the CB1 antagonist AM251 (10 µM, 1 h) significantly increased both evoked (CTRL: 99.50 ± 7.8 pA, AM251: 118.8 ± 3.4 pA; n = 6, *p* = 0.04; Figure 2B) and spontaneous glutamatergic activity (CTRL: 4.37 ± 0.44 Hz, AM251: 5.4 ± 0.26 Hz; n = 6, *p* = 0.02) (Figure 2D). On the contrary, bath application of the CB1 agonist ACEA (10 µM, 1 h) caused a clear reduction in both evoked (Figure 2C) and spontaneous EPSCs (CTRL: 95.14 ± 3.9 pA, ACEA: 76.86 ± 4.5 pA, n = 7, *p* = 0.02; CTRL: 4.63 ± 0.38 Hz, ACEA: 3.61 ± 0.35 Hz, n = 7, *p* = 0.02; Figure 2E). Thus, modulation of EC signaling tonically regulated excitatory synapses in our ex vivo hippocampal system. To validate this data, a new set of electrophysiological experiments was conducted. Single-neuron recordings were performed, and the changes following the application of AM251 and ACEA were observed over the subsequent 25–30 min of spontaneous glutamatergic synaptic activity. At least five minutes baseline were recorded before starting pharmacological approach (ACEA baseline 3.9 ± 0.3 over 25 min drug 3.2 ± 0.3 *p* = 0.21 n = 3; AM251 baseline 4.06 ± 0.44 over 25 min drug 5.2 ± 0.2 *p* = 0.01 n = 3) (Figure 2F). The data demonstrated that, even in this experimental configuration, ACEA induced a clear reduction in synaptic activity, while the application of AM251 determined a significant increase in the frequency of spontaneous glutamatergic activity (Figure 2G).

It has been reported that CB1 modulates various cellular signaling pathways, among which the phosphatidylinositol 3-kinase (PI3K)/Akt and the mitogen-activated protein kinases (MAPK) pathways. In mouse hippocampus CB1 activation has been related to neuroprotective effect through Akt phosphorylation [39,40]. To verify the CB1-dependent activation of this intracellular pathway, we used hippocampal neurons isolated from C57BL/6 E17.5 mice and cultured in vitro for 7 days in a neuron-specific media to obtain a 90% neuronal culture and to allow the appearance of extended axons and the establishment of synaptic connections among neurons. We were confident that our primary mouse neurons cultured up to 7 days were similar to the hippocampus by the first month in terms of brain developmental stage, the time at which brain enters a mature stage, synaptic formations are established and connections are active [41,42].

The culture of hippocampal neurons was treated with 1 µM ACEA at different times, and Akt phosphorylation was analyzed on protein extracts by Western blotting. We found that ACEA increased Akt phosphorylation in a time-dependent manner in hippocampal neurons and that pretreatment with the CB1 antagonist AM251 abolished this effect (Figure 3A,B), suggesting that CB1 was specifically involved in the activation of this pathway.

### 3.2. Expression of Nrxns Splice Variants Is Modulated by CB1 Receptors Signaling

Next, we analyzed the expression of *Nrxn*1–3 SS4 splice variants in hippocampal slices in basal conditions and after perfusion in continuous presence of agonist/antagonist for 6 h. RT-PCR analysis showed that in basal conditions, the *Nrxn1* splice variants, which include (SS4+) and exclude (SS4−) the exon, were expressed at similar levels, while the skipped SS4− variant was mainly expressed for *Nrxn2* and *Nrxn3* (Figure 4A). Interestingly, treatment with the CB1-targeting drugs modulated AS of *Nrxns1*–*3*: AM251 induced a significant increase in SS4 exon inclusion (SS4+) in *Nrxn2, Nrxn3* and, albeit to a lesser extent, also in *Nrxn1* (Figure 4B); in contrast, the CB1 agonist ACEA mildly induced SS4 skipping in *Nrxn1* and *Nrxn2* promoting the expression of the SS4− variant, whereas no significant effect was observed in *Nrxn3* (Figure 4B). Selective primers for α or β *Nrxns* isoforms were also used to investigate the modulation of the splicing specifically in these isoforms. In line, AM251 treatment significantly increased the SS4^+^ variants in both types of isoforms for the three *Nrxn* genes, although the effects were statistically more robust for the αNrxns (Appendix A). These results suggest that pharmacologic modulation of CB1 activity in the hippocampus regulated SS4 splicing in the *Nrxn1*–*3* genes. Notably, splicing of the SS4+ variants was associated with increased neurotransmission (eEPSCs and sEPSCs), while the SS4− variants with reduced neurotransmission.

### 3.3. SLM2 Implication in Nrxns Splicing Pattern at SS4 Site

Alternative splicing of *Nrxns* at SS4 is primarily mediated by the action of the STAR family of RNA-binding proteins [19,43,44,45,46,47,48]. We recently found that the STAR protein SLM2 is a main regulator of SS4 splicing in the adult cortex [21,45]. SLM2 is also highly expressed in the CA1–3 regions of the hippocampus [44]. Thus, we set out to investigate its role in the EC-mediated alternative splicing of *Nrxns* in the adult hippocampus in *Slm2* ko mice. To this end, we treated hippocampal slices from wild type and *Slm2* ko mice [43] with vehicle (DMSO) as control, and either ACEA or AM251 (10 µM) drugs for 6 h, and then the SS4 splice variants expression of *Nrxns* were evaluated by RT-PCR analysis. We showed that *Slm2* knockout mice displayed virtually complete inclusion of the SS4 exon in all three *Nrxn* transcripts (Figure 5A) and, more interestingly, modulation of the splicing event at SS4 by AM251 or ACEA was completely suppressed in the absence of *Slm2* (Figure 5A,B). We also showed by RT-qPCR that the *SLM2* expression level was not altered after drug treatments (Figure 5C). Thus, the lack of SLM2 abolished the modulation exerted by the ECS of *Nrxn1*–*3* alternative splicing and constitutively imposed the SS4+ pattern induced by stimulation of CB1 receptors activity in the wild-type context.

Next, we analyzed the effect of *Slm2* ablation on the hippocampal neuronal circuitry by recording eEPSCs and sEPSCs in the subiculum. Untreated *Slm2* ko hippocampal slices showed a significant reduction in glutamatergic amplitude of the events triggered by electrical stimulation of the CA3 region (*Slm2* ko: 88.57 ± 6.9 pA; WT: 123.3 ± 7.1 pA n = 7, *p* = 0.005) (Figure 6A). Accordingly, the spontaneous excitatory events were also significantly reduced in *Slm2* ko slices with respect to hippocampal slices isolated from WT mice (*Slm2* ko: 2.69 ± 0.24 Hz; WT: 4.2 ± 0.36 Hz n = 7, *p* = 0.03) (Figure 6B). Since the effects of *Slm2* ablation mimicked those of ACEA on hippocampal neurotransmission, we tested whether SLM2 expression is required for the response to pharmacological modulation of CB1 activity. Interestingly, treatment of *Slm2* ko hippocampal slices with either AM251 (10 µM) or ACEA (10 µM) did not induce statistically significant changes in both evoked glutamatergic events (eEPSCs: untreated 90.25 ± 6.2; Slm2 ko AM251 98.13 ± 3.88 pA, n = 7 *p* = 0.14 Slm2 ACEA 88.60 ± 7.8 n = 5 *p* = 0.87) (Figure 6C) and spontaneous (untreated 2.62 ± 0.24; *Slm2* ko AM251 3.20 ± 0.23 pA *p* = 0.11; *Slm2* ko ACEA 2.90 ± 0.25; *p* = 0.45, n = 7) (Figure 6D), indicating that SLM2 expression and *Nrxn1*–*3* splicing are required for the modulation of neurotransmission by CB1-dependent signaling in the hippocampus. Similarly, single-neuron recordings with the application of ACEA and AM251 for over 25 min confirm the ineffectiveness of these two drugs in Slm2KO mice (Figure 6E,F) (AM251 baseline 2.1 ± 0.44 Hz over 25 min drug application 2.9 ± 0.26 Hz n = 3 *p* = 0.93; ACEA baseline 1.9 ± 0.42, over 25 min drug application 2.1 ± 0.29 Hz n = 3 *p* = 0.66).

The effects of *Slm2* depletion were also investigated on the inhibitory circuit by recording spontaneous and evoked GABAergic (eIPSCs and sIPSCs, respectively) events in the subiculum. To avoid any contamination, glutamatergic input was suppressed by adding specific glutamate receptor antagonists, such as CNQX (10 µM) and APV (10 µM). This analysis indicated that hippocampal slices from *Slm2* ko mice showed a significant reduction in both spontaneous (*Slm2* ko: 2.80 ± 0.34 Hz; WT: 4.25 ± 0.27 Hz; n = 6, *p* = 0.01) and evoked (*Slm2* ko: 40.17 ± 4.4 pA; WT: 57.83 ± 5.59 pA; n = 6, *p* = 0.04) inhibitory GABAergic currents (Appendix A).

## 4. Discussion

This study reports novel findings on the relationship between CB1 signaling, neuronal activity and expression of specific neurexin splice variants in the hippocampus. Our data reveals a feedback mechanism of regulation of CB1-dependent neurotransmission involving modulation of alternative splicing of neurexins. The identification of this interplay could help to understand how the endocannabinoid system plays a pivotal role in the fine-tuned regulation of neural transmission and in the maintenance of brain homeostasis.

We employed ex vivo hippocampal slices pharmacologically treated with an agonist or antagonist of CB1, and we investigated the excitatory output of CA1 pyramidal neurons onto subiculum neurons, which are considered the major efferent pathway of the hippocampus [49]. In the hippocampus, CB1 is highly expressed by neurons at pre-synaptic terminals and, when activated, leads to inhibitory retrograde signaling and suppression of pre-synaptic neurotransmitter release [32,50,51]. First of all, our data indicate the presence of a functionally active CB1 receptor in the hippocampus. We demonstrated that pharmacological stimulation with agonist/antagonist of CB1 in hippocampal slices modulated neurotransmission. In addition, we found that treatment of isolated hippocampal neurons with a CB1-selective agonist ACEA activated the PI3K pathway by increasing Akt phosphorylation in a way attenuated by a CB1-selective antagonist AM251. Then, in our ex vivo system, we evaluated the possible correlation between CB1-dependent neural activity and alternative splicing of neurexins at SS4. Alternative splicing is the process by which many splicing variants can be produced. Alternative splicing of *Nrxns1*–*3* yields a plethora of variants, each with a specific binding to different post-synaptic ligands. Through the expression of different isoforms and splicing variants, neurexins contribute to control synapse properties and increase synaptic plasticity [14]. We showed that the Nrxn1–3 splice variants skipping the SS4 exon (SS4− variants) are predominantly expressed in the hippocampus under basal conditions. Treatment with the CB1 antagonist AM251 inhibited CB1 signaling and increased glutamatergic activity. This change in neural activity correlated with a switch in the expression of splice variants of Nrxns. Indeed, AM251 treatment promoted the inclusion of the SS4 exon in all three neurexins, increasing the expression of the SS4+ variants of Nrxns. By contrast, treatment with the CB1 agonist ACEA activated the CB1 signaling and caused a decrease in both evoked and spontaneous EPSCs. This effect on neural activity was associated with a further increase in the expression of SS4− splice variants, mainly in Nrxn1–2. Thus, our findings indicated that CB1-dependent neural activity was coupled to a modulation of the alternative splicing of Nrxn 1–3 at the SS4 site. Several studies have already reported that alternative splicing of neurexins is activity-dependent [19,37,52,53,54,55]. It has been shown that SS4 alternative splicing of pre-synaptic Nrxn1 and Nrxn3 controls the receptor composition of excitatory hippocampal synapses. In particular, the Nrxn1 SS4+ isoform increases NMDA receptor responses, whereas Nrxn3 SS4+ suppresses AMPA receptor responses, indicating that pre-synaptic neurexin splicing shapes the postsynaptic glutamate receptor composition [56,57].

A previous study suggested an interplay between the ECS and a specific neurexins isoform in the regulation of neural circuits. This study highlighted a specific role of the SS4− splice variant of β-neurexins in the hippocampus in the control of synaptic strength at excitatory synapses. Notably, the SS4− splice variant, and not the SS4+, induced a decrease in postsynaptic 2-arachidonoylglycerol synthesis, thus promoting neurotransmission [36]. Following this evidence, we herein speculate that the CB1-dependent neurotransmission could promote the expression of specific Nrxn splice variants whose function could be important in restoring basal synaptic transmission and thus maintaining synaptic homeostasis. Indeed, we found that, in the presence of ACEA, neurotransmission decreased and, concomitantly, overexpression of the SS4− splice variants occurred. As demonstrated by Anderson et al., this splice variant decreases postsynaptic EC synthesis and promotes neurotransmission to restore the basal tone of neural activity. Conversely, we found that the inhibition of CB1 signaling by AM251 promoted neurotransmission and determined alternative splicing of Nrxns towards the SS4+ variants, which should be important in stabilizing synaptic transmission by lowering neural activity. Our model of preservation of normal synaptic transmission, following stimulation of CB1, through alternative splicing mechanism of pre-synaptic molecules is shown in Figure 7.

These findings suggest a link between CB1-dependent neurotransmission and the expression of alternatively spliced isoforms of Nrxns in the hippocampus with opposite roles in stabilizing synaptic transmission. Compensatory adjustments have been described in neural circuits to restore the system to its basal function when synaptic activity is increased (or decreased) above a basal level [58]. Future studies will be required to better elucidate the specific role played by the neurexin SS4+ splice variants of Nrxns in maintaining optimal synaptic transmission.

To confirm the link between CB1-dependent neurotransmission and the expression of Nrxns splice variants, we used Slm2-null mice. SLM2 is the splicing factor involved in the alternative splicing of Nrxns at SS4. We showed that in Slm2-null mice, alternative splicing at SS4 was suppressed, and the variant SS4+ was the only expressed in the hippocampus. We reported that, in this condition, the pharmacological regulation of CB1-dependent neural activity was blocked, indicating that Nrxn1–3 alternative splicing at SS4 is crucial for the proper response of the CA1-subiculum circuit to EC signaling. Moreover, this finding uncovers a role played by SLM2 in this hippocampal circuit. In line with this new evidence, previous data reported that the RNA binding protein SLM2 is highly expressed in glutamatergic CA1 in the mouse hippocampus and that most SLM2-bound mRNAs encode for synaptic proteins, suggesting a role in the synaptic functions and plasticity [57].

Moreover, our results open new perspectives on the role of the ECS–neurexin axis in the modulation of learning and memory. Synaptic plasticity—the fundamental substrate of memory formation—relies on activity-dependent mechanisms capable of adapting synaptic properties to changing functional demands. We show that CB1 receptor signaling dynamically regulates the alternative splicing of the SS4 exon in Nrxn1–3, thereby producing splice isoforms that may differentially shape synaptic efficacy. The CB1-induced shift toward SS4− variants, which suppress endocannabinoid synthesis and promote neurotransmission, may function to restore synaptic homeostasis during periods of reduced excitability. Conversely, CB1 inhibition favors SS4+ inclusion, potentially acting as a brake to counteract excessive excitatory drive. Given the central role of the hippocampus in memory encoding, this fine-tuned, activity-dependent splicing program may constitute a critical molecular interface linking synaptic activity to gene expression and memory-related plasticity [22,23,24].

While our findings provide novel insights into the modulation of neurexin alternative splicing by CB1 receptor signaling, some limitations must be acknowledged. Most notably, the experiments were conducted using ex vivo hippocampal slices, which, although preserving some native connectivity and architecture, do not fully recapitulate the complexity of the intact brain. In vivo systems involve intricate interactions among multiple brain regions, neuromodulatory inputs, vascular dynamics and behavioral states that cannot be entirely reproduced ex vivo. Therefore, it is possible that the extent and dynamics of CB1-dependent modulation of neurexin splicing—and its downstream effects on synaptic transmission—could differ in an intact organism. Further in vivo studies will be necessary to validate whether the observed mechanisms contribute similarly to neuronal function and synaptic plasticity under physiological or pathological conditions.

In conclusion, our data reveals a novel interplay between the ECS and neurexin alternative splicing in the hippocampus and suggests a potential mechanism by which ECS plays a critical neuromodulator role by ensuring a fine regulation of neuronal activity through the modulation of alternative splicing of presynaptic proteins neurexins. These findings may contribute to a better understanding of certain hippocampal stress and learning disorders, such as depression, in which altered homeostatic mechanisms regulating synaptic plasticity need to be restored [57,58].

## Figures and Tables

**Figure 1 cells-14-00972-f001:**
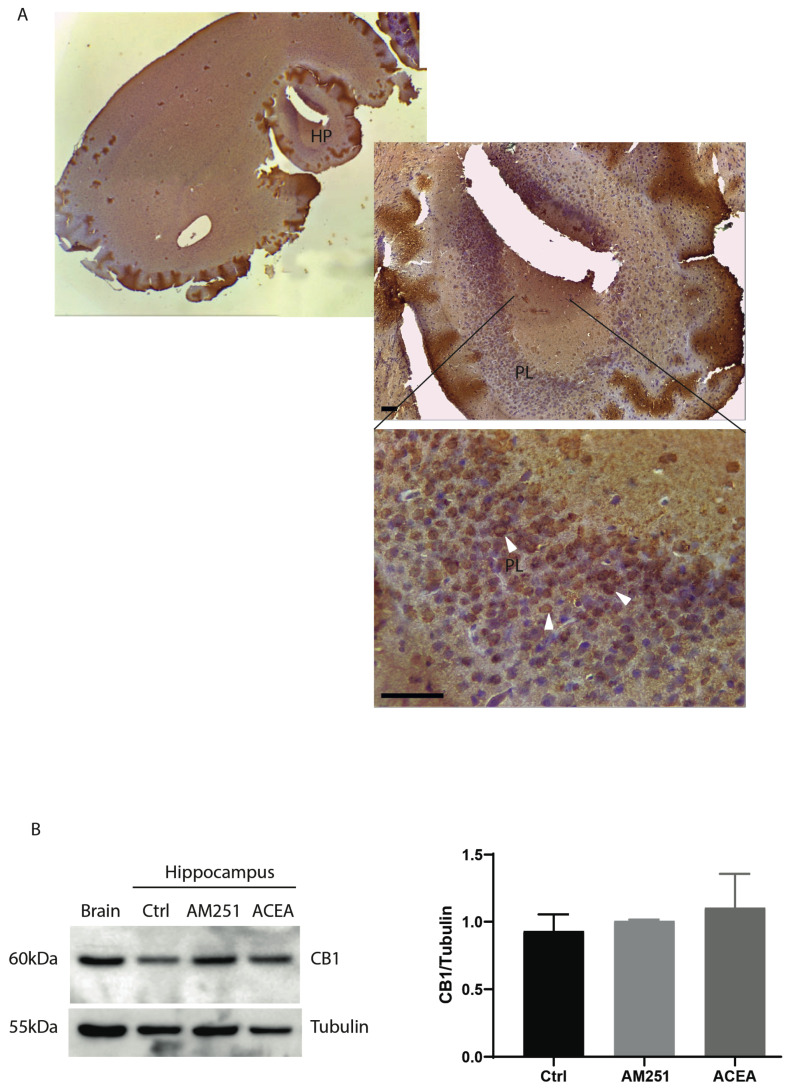
Expression CB1 receptors in hippocampus (HP). (**A**) Parasagittal hippocampal section of 5 μm from one-month-old mice stained with anti-CB1 antibody. General view of CB1-immunoreactivity on hippocampus. The bottom panel shows the highest magnification of the pyramidal layer of a parasagittal section of ventral hippocampus (magnification 40×, scale bar 25 µm). PL: Pyramidal cell layer. CB1-positive cells are indicated by arrowheads. (**B**) Western blotting with anti-CB1 antibody on protein extracts from hippocampal slices treated for 6 h with ACEA (10 μM) or AM251 (10 μM). Whole brain is used as CB1-positive control. On the right, histograms reporting the densitometric analysis (Image J-win64). Tubulin is used as reference for protein loading. Results were obtained from n = 3 independent experiments. Error bars represent SD. Columns were compared with an unpaired *t*-test. No statistical differences were observed.

**Figure 2 cells-14-00972-f002:**
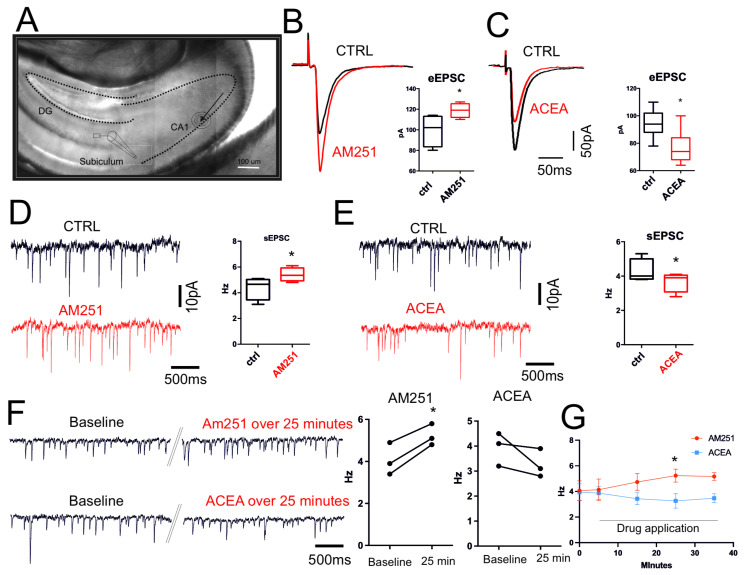
CB1-dependent neurotransmission in ex vivo hippocampus. (**A**) IR-image of horizontal hippocampal slice used for electrophysiological recordings reporting the place of recordings as well as site of electrical stimulation. (**B**) Representative traces of evoked EPSCs of subiculum neurons recorded before (black) and after (red) incubation with AM251 (10 µM, 1 h) (*p* = 0.04). (**C**) Representative traces of evoked EPSCs of subiculum neurons recorded before (black) and after (red) incubation with ACEA (10 µM, 15 min) (*p* = 0.02). (**D**) Representative traces of spontaneous EPSCs recorded before (black) and after (red) incubation with AM251 (10 µM, 1 h) (*p* = 0.02). (**E**) Representative traces of spontaneous EPSCs recorded before (black) and after (red) incubation with ACEA (10 µM, 1 h) (*p* = 0.02). Box plots show that both AM251 and ACEA significantly affect both evoked EPSC amplitude and spontaneous EPSC frequency. Results were obtained from n = 3 mice for AM251, n = 3 mice for ACEA. For each mouse, 4 different hippocampal slices were used. (**F**) Representative traces of spontaneous EPSCs recorded from single neuron. Traces show event at beginning of the recording (Baseline) and after continuous application of AM251 or ACEA. Double oblique line marks where the single trace has been cut for representation purposes. (**G**) Plot shows the frequency changes over time triggered by bath application of ACEA and AM251. Statistical analysis was performed with paired or unpaired tests. Error bars represent SEM. * *p* < 0.05.

**Figure 3 cells-14-00972-f003:**
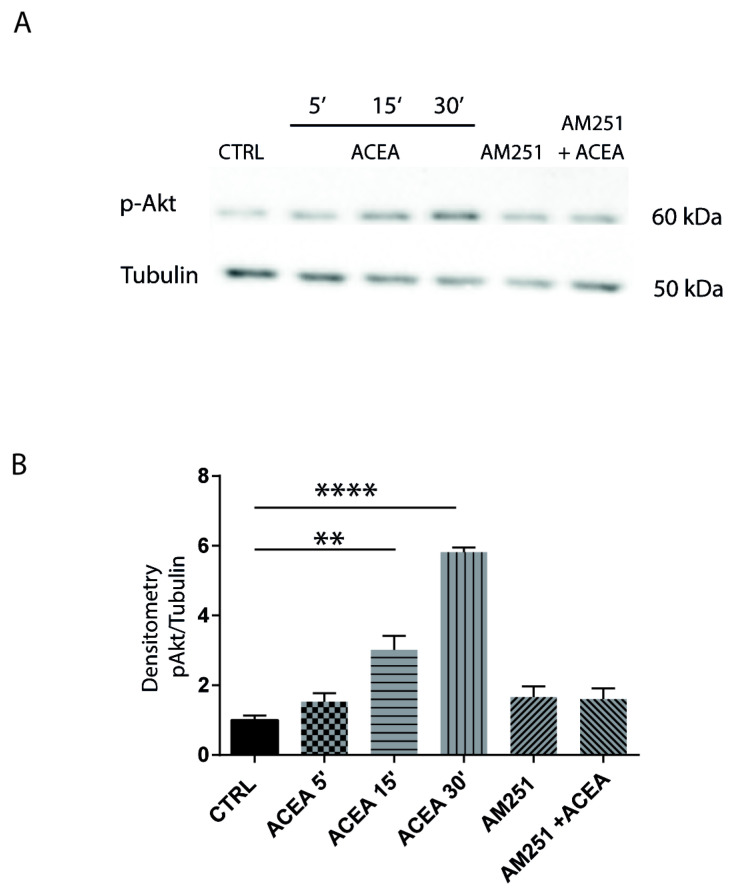
CB1-dependent intracellular signaling pathways in hippocampal neurons. (**A**) Western blotting showing the time course analysis of Akt phosphorylation in hippocampal neurons treated with ACEA, (5′, 15′, 30′) or with AM251 alone and in combination with ACEA (40′ before agonist). Tubulin is used as reference for protein loading. (**B**) Histogram reports the densitometric analysis of Western blotting. CB1 agonist ACEA induces Akt phosphorylation, and the effect is blocked by the antagonist AM251. (CTRL-ACEA 15′ *p* = 0.0045 **; CTRL-ACEA 30′ *p* < 0.0001 ****). Results were obtained from n = 3 independent experiments. Columns were compared with one-way ANOVA test, followed by Bonferroni test.

**Figure 4 cells-14-00972-f004:**
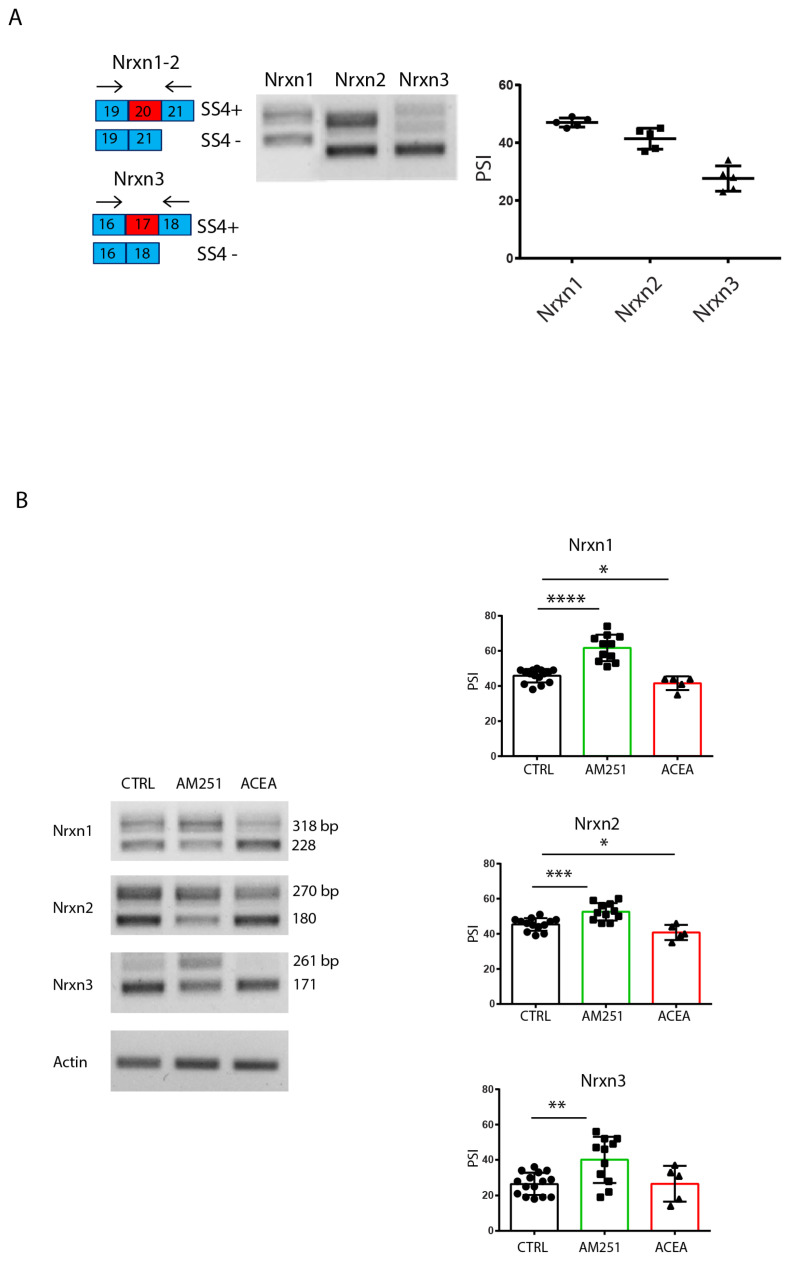
Stimulation of CB1 receptors modulates Nrxns splicing pattern at SS4 in hippocampal slices. (**A**) RT-PCR analysis of SS4 splice variants of Nrxns1–3 expression in hippocampal slices from one-month-old mice. Scheme of SS4 splice variants showing the selective primers flanking the alternative exon 20 for Nrxn1–2 and exon 17 for Nrxn3. On the right, the relative scatter plot showing the PSI, calculated by densitometric analysis (Image Lab 6.0) of Nrxns1–3 splicing pattern from hippocampal slices. Results were obtained from n = 5 independent experiments (n = 5 mice). Error bars represent SD. (**B**) RT-PCR analysis of SS4 splice variants of Nrxns1–3 expression in hippocampal slices, in basal condition and after stimulation for 6 h with ACEA (10 μM) and AM251 (10 μM). On the right, the relative scatter plots showing the PSI, calculated by densitometric analysis (Image Lab 6.0) of Nrxns1–3 splicing pattern from hippocampal slices. Results were obtained from n = 3 mice for AM251 and n = 3 mice for ACEA. For each mouse, 4 different hippocampal slices were used. Statistical analysis was performed with unpaired *t* test (Nrxn1: CTRL-AM251 *p* < 0.0001 ****, CTRL-ACEA *p* = 0.0451 *, Nrxn2: CTRL-AM251 *p* = 0.0005 ***, CTRL-ACEA *p* = 0.0354 *, Nrxn3: CTRL-AM251 *p* = 0.0017 **). Error bars represent SD.

**Figure 5 cells-14-00972-f005:**
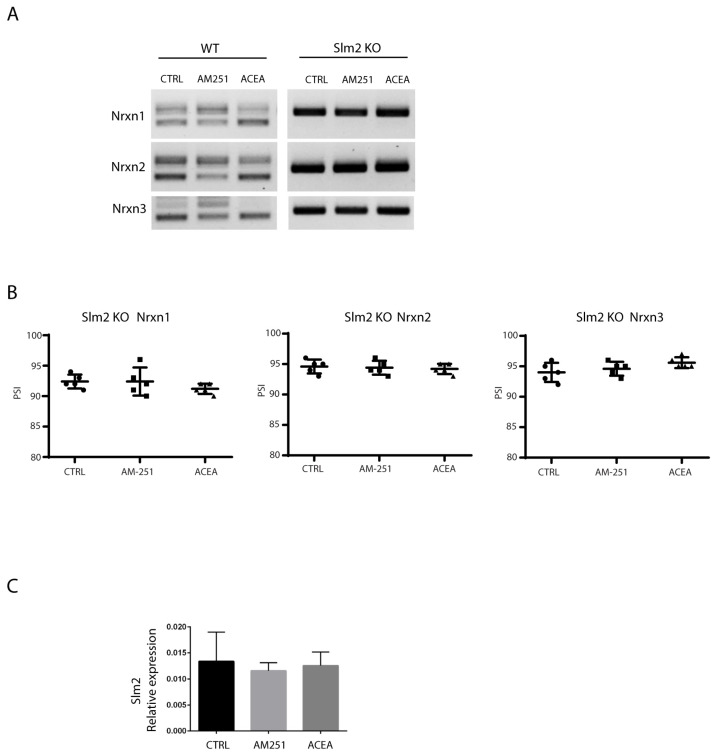
Nrxns splicing pattern in hippocampal slices from WT and Slm2 ko mice. (**A**) RT-PCR analysis performed with selective primers to amplify SS4 splice variants of Nrxns1–3 in hippocampal slices from WT and *Slm2* ko mice after stimulation for 6 h with ACEA or AM251 (10 μM). In ko mice, *Slm2* absence prevents Nrxns skipping towards the SS4− isoform, and treatments with the drugs are not able to modulate this pattern. (**B**) Relative scatter plots showing the PSI, calculated by densitometric analysis (Image Lab 6.0) of Nrxns1–3 splicing pattern in hippocampal slices from *Slm2* ko mice. (**C**) qRT-PCR analysis of Slm2 on hippocampal slices from WT mice treated with ACEA or AM251 (10 μM). Results were obtained from n = 3 independent experiments using n = 2 mice. For each mouse, 4 hippocampal slices were used. Error bars represent SD. Statistical analysis was performed by unpaired *t*-test. No statistical differences were detected.

**Figure 6 cells-14-00972-f006:**
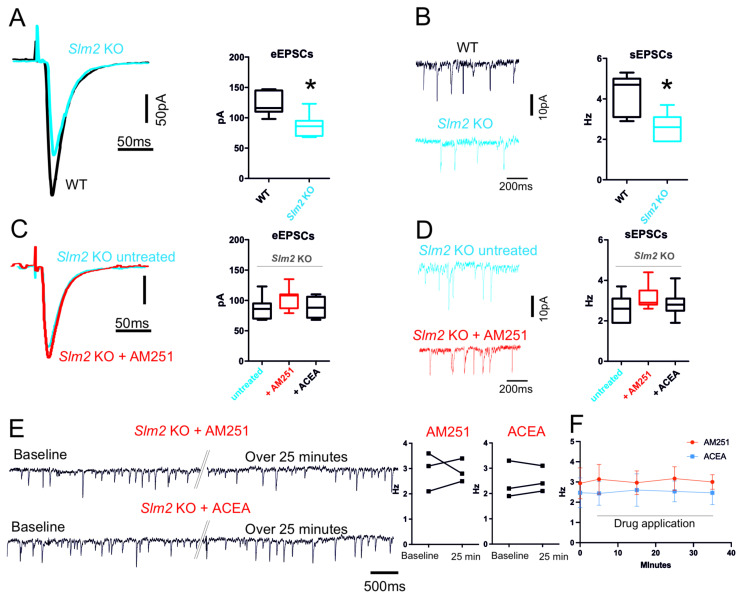
*Slm2* ko electrophysiological characterization of glutamatergic input. (**A**) Representative traces of evoked EPSCs of subiculum neurons recorded from WT (black) and *Slm2* ko mice (blue) (*p* = 0.005). (**B**) Representative traces of spontaneous EPSCs recorded from WT (black) and *Slm2* ko mice (blue) (*p* = 0.03). Box plots show that *Slm2* ko mice have a significant reduction in both evoked EPSCs amplitude and spontaneous EPSCs frequency. (**C**) Representative traces of evoked EPSCs of subiculum neurons of *Slm2* ko mice recorded before (blue) and after (red) incubation with AM251 (10 µM). (**D**) Representative traces of spontaneous EPSCs before (blue) and after (red) incubation with AM251 (10 µM). Box plots summarize the effect of AM251 incubation as well as of ACEA in *Slm2* ko mice. (**E**) Representative traces of spontaneous EPSCs recorded from single neuron of Slm2 KO mice. Traces show event at beginning of the recording (Baseline) and after continuous application of AM251 or ACEA (>25 min). Double oblique line marks where the single trace has been cut for representation purposes. (**F**) Plot shows the frequency changes over time triggered by bath application of ACEA and AM251. Statistical analysis was performed with paired and/or unpaired *t* test. Error bars represent SEM. * *p* < 0.05.

**Figure 7 cells-14-00972-f007:**
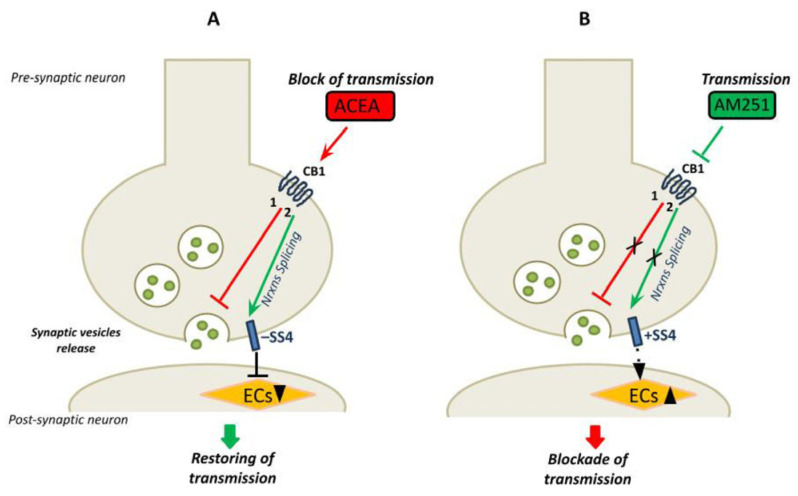
The model shows the link between CB1 receptors signaling, neurotransmission and the neurexin SS4 splice variants expression in mouse hippocampus. (**A**) The CB1 receptors specific agonist ACEA activates the CB1 receptors leading to: (1) inhibition of the synaptic vesicles release and block of neurotransmission (red line); (2) alternative splicing of Nrxns and main expression of Nrxns splice variants SS4− (green arrow). The SS4− reduces the synthesis of en-docannabinoids (ECs) at postsynaptic neurons, restoring neurotransmission [36]. (**B**) The CB1 receptors specific an-tagonist AM251 blocks the receptor signaling leading to: (1) promotion of vesicles release and neurotransmission (red line) and (2) inhibition of alternative splicing of Nrxns, and expression of the SS4+ splice variants (green arrow). The SS4+ may play a role in reducing the neurotransmission to basal level (by inducing the synthesis of ECs). In the proposed model the expression of Nrxn splice variants at SS4, associated with changes in neurotransmission, may function in maintaining the basal neural activity.

**Table 1 cells-14-00972-t001:** List of primer sequences.

Nrxns SS4 Splice Variants [19]	
Nrxn1 (+/−SS4) 318–228 bp	Fw: 5′-TGT TGG GAC AGA TGA CAT CGC C-3′
	Rv: 5′-GAG AGC TGG CCC TGG AAG GG-3′
Nrxn2 (+/−SS4) 270–180 bp	Fw: 5′-GTG CGC TTT ACT CGA AGT GGT G-3′
	Rv: 5′-CCC ATT GTA GTA GAG GCC GGA C-3′
Nrxn3 (+/−SS4) 261–171 bp	Fw: 5′-TTG TGC GCT TCA CCA GGA ATG-3′
	Rv: 5′-AGA GCC CAG AGA GTT GAC CTT G-3′
Nrxns Alpha/Beta [38]	
Nrxn1 (+/−SS4) α: 645–555 bp β: 699–609 bp	Fw (α): 5′-CAG CAC AAC CTG CCA AGA-3′
	Fw (β): 5′-CCT GGC CCT GAT CTG GAT AGT-3′
	Rv (αβ): 5′-GAG AGC TGG CCC TGG AAG GG-3′
Nrxn2 (+/−SS4) α: 661–571 bp β: 597–507 bp	Fw (α): 5′-CAC CAC CTG CAC CGA AGA G-3′
	Fw (β): 5′-GTG CCC ATC GCC ATC AA-3′
	Rv (αβ): 5′-CCC ATT GTA GTA GAG GCC GGA C-3′
Nrxn3 (+/−SS4) α: 584–494 bp β: 598–508 bp	Fw (α): 5′-CTG TGA CTGCTC CAT GAC ATC ATATT-3′
	Fw (β): 5′-AAGCACCACTCTGTGCCTATTTCT-3′
	Rv (αβ): 5′-AGA GCC CAG AGA GTT GAC CTT G-3′
β-Actin 135 bp	Fw: 5′-CTG TCG AGT CGC GTC CAC-3′
	Rv: 5′-GCT TTG CAC ATG CCG GAG-3′
Slm2 71bp	Fw: 5′-TGATGGCGGAGAAGGACTCT-3′
	Rv: 5′-TTCTATTTCTCGGTTCACCAAGCG-3′

## Data Availability

The original contributions presented in this study are included in the article/Appendix A. Further inquiries can be directed to the corresponding author.

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
