# Peer review of "Modulation of Neurexins Alternative Splicing by Cannabinoid Receptors 1 (CB1) Signaling"

_cells, 2025, doi:10.3390/cells14130972_

Round 1

Reviewer 1 Report

Comments and Suggestions for Authors

Comments are listed here for the author’s consideration to further improve the quality and overall impact of the manuscript.

English edition is necessary, as there are some errors in the manuscript.

Modify the title: “Modulation of Neurexins Alternative Splicing by cannabinoid receptor 1 (CB1) Signaling”

In the conclusion of the summary the authors mention: “We propose that the finetuned regulation of Nrxn1-3 genes alternative splicing may play an important role in the feedback control of neurotransmission by the endocannabinoid system and may also contribute to memory related synaptic plasticity.” It is essential to note that authors cannot assert a link with memory-related synaptic plasticity, as there is currently no direct evidence supporting this in the present study. Modify your conclusion.

Line 39-40. “Synaptic modifications of neural connections underlying the cognitive basis of learning and memory.” It is unclear what the intended message is. Correct it.

Line 93-95. “…whose functions should be critical in the preservation of suggesting that alternative splicing of Nrxns could be a mechanism by which endocannabinoids exert a fine-tuned regulation of neurotrasmission in the hippocampus.” It is unclear what the intended message is. Correct it.

Line 98-102. Include it as another section such as Animals.

Could you explain why the neuronal cultures were treated with ACEA (1μM) or AM251(2 μM) for 16 hours?

Why horizontal hippocampal slices were treated with drugs ACEA and AM251 for 1 hour for electrophysiological recordings? And why for 6 hours for Nrxns molecular analyzes? Indicate it.

Authors mention at Figure 2 that A) IR-image of coronal hippocampal slice…. But in the methods section the authors indicate that they were used horizontal hippocampal slices (300 μm). Could you clarify whether the slices are coronal or horizontal? Please correct it.

Why total RNA was extracted from hippocampus isolated from C57BL/6 male mice at E17.5 age?  And why also from hippocampal slices from young male mice of 1 month? Are there differences in the methodology or expression of RNA at different ages?  Indicate it.

Define what it is the percentage of splicing inclusion (PSI).

At 2.5. Western Blotting section, specify which hippocampal tissues were analyzed with the technique. Were they protein extracts from hippocampal neurons, hippocampal slices, hippocampus?  Indicate it.

At 2.6. section specify which hippocampal tissue was analyzed with the CB1 immunohistochemistry technique. The concept of using sections from the hippocampus is not without its ambiguities. In figure 1 authors indicated: A) Parasagittal hippocampal section from one month old WT mice. What tissue was used in this technique? Why sections were then counterstained with hematoxylin? Indicate it.

Figure 1. A) Enhance the image. Indicate with arrows which cells express CB1 receptors. Include a microphotograph of the entire slice. B) What does the first column mean? Brain? What does that band correspond to? Authors mention: Tubulin is used as reference gene. As reference gene? Or protein?  Indicate it.

At figure 3: A) Time course analysis of Akt phosphorylation in hippocampal neurons treated with ACEA (5’, 15’, 30’) or with AM251 alone and in combination with ACEA (40’ before agonist). However, at section 2.1. Primary hippocampal neuronal cultures authors mention that hippocampal neuronal cultures were treated with vehicle (DMSO) and drugs ACEA (1μM) or AM251(2 μM) for 16 hours. At what time were the hippocampal neuron cultures actually treated? 16 hours or 5’, 15’, 30’ and 40’ before agonist? Explain and indicate it correctly. Also at Figure 3 section B is missing. Authors mention “Tubulin is used as normalizer”, What it means normalizer? Correct it.

Why were different drug administration times used (1 or 6 hours, 5-15-30-40 minutes, 16 hours? Justify your answer.

Figure 4. Correct in B) NRXS 2 and 3 graphs, the numbers on the y axis.

Line 441-443. Please include the figure 7. It was inadvertently omitted in the writing.

Authors do not discuss anything regarding the Akt phosphorylation in hippocampal neurons. Include the information in the discussion section.

Additionally, authors should discuss the limitations of their study, which were determined using ex vivo hippocampal slices. It is important to note that the results obtained in this study could vary when using an in vivo model with an intact animal.

Comments on the Quality of English Language

The English edition is necessary because there are some errors in the manuscript.

Author Response

We would like to thank the reviewers for the suggestions provided to improve the work, below are our responses to the points raised by the reviewers

REF 1

Comments are listed here for the author’s consideration to further improve the quality and overall impact of the manuscript.

English edition is necessary, as there are some errors in the manuscript.

Text has been carefully revisioned and typos and/or errors have been corrected

Modify the title: “Modulation of Neurexins Alternative Splicing by cannabinoid receptor 1 (CB1) Signaling”

As suggested by the reviewer the title has been changed.

In the conclusion of the summary the authors mention: “We propose that the finetuned regulation of Nrxn1-3 genes alternative splicing may play an important role in the feedback control of neurotransmission by the endocannabinoid system and may also contribute to memory related synaptic plasticity.” It is essential to note that authors cannot assert a link with memory-related synaptic plasticity, as there is currently no direct evidence supporting this in the present study. Modify your conclusion.

We thank the reviewer for this comment. Accordingly the conclusion of the summary has been modified.

Line 39-40. “Synaptic modifications of neural connections underlying the cognitive basis of learning and memory.” It is unclear what the intended message is. Correct it.

We corrected line 39-40 and now we state: “Synaptic modifications of neural connections underlie the cognitive basis of learning and memory”.

Line 93-95. “…whose functions should be critical in the preservation of suggesting that alternative splicing of Nrxns could be a mechanism by which endocannabinoids exert a fine-tuned regulation of neurotrasmission in the hippocampus.” It is unclear what the intended message is. Correct it.

We corrected line 93-95 and now we state: Deletion of the involved splicing factor Slm2 abolished the SS4 alternative splicing of Nrxns and concomitantly it suppressed the CB1-dependent neural activity, suggesting that alternative splicing of Nrxns could be a mechanism by which endocannabinoids exert a fine-tuned regulation of neuro-trasmission in the hippocampus.

Line 98-102. Include it as another section such as Animals.

As suggested we introduced a new section 2.1 “Animals”.

Could you explain why the neuronal cultures were treated with ACEA (1μM) or AM251(2 μM) for 16 hours?

We thank the reviewer for this right observation. We are sorry for this mistake. We have never performed a 16-hour treatment in our experiments on neuronal cultures. Figure 3 shows the experiment performed on cultured hippocampal neurons to study the activated intracellular pathway. The relative legend reports the time course analysis of Akt phosphorylation in hippocampal neurons treated with ACEA, (5’, 15’, 30’) or with AM251 alone and in combination with ACEA (40’ before agonist). We corrected this in the text.

Why horizontal hippocampal slices were treated with drugs ACEA and AM251 for 1 hour for. electrophysiological recordings? And why for 6 hours for Nrxns molecular analyzes? Indicate it.

Slices used for electrophysiology cannot tolerate a 6-hour treatment period; therefore, a shorter treatment time was used to better preserve the neural vitality of the slices during the functional recordings. After preliminary investigation, this shorter time (1 hour) was considered sufficient to trigger and observe the acute drug effect on synaptic transmission. For better explanation of drugs timing rationale please refere to the below answer: “Why were different drug administration times used (1 or 6 hours, 5-15-30-40 minutes, 16 hours? Justify your answer”.

Authors mention at Figure 2 that A) IR-image of coronal hippocampal slice…. But in the methods section the authors indicate that they were used horizontal hippocampal slices (300 μm). Could you clarify whether the slices are coronal or horizontal? Please correct it.

We apologize for the error. Slices are horizontal. Text have been changed accordingly.

Why total RNA was extracted from hippocampus isolated from C57BL/6 male mice at E17.5 age?  And why also from hippocampal slices from young male mice of 1 month? Are there differences in the methodology or expression of RNA at different ages?  Indicate it.

We thank the reviewer for this observation. We have now corrected this point in the text clarifying in the new section 2.5 that in all our PCR experiments we only used RNA extracted from hippocampal slices of young male mice of 1 month. This is reported in the relative Figure legends.   Hippocampal neurons from C57BL/6 male mice at E17.5 age. were not used to extract RNA but were used to prepare protein extracts to analyzed in western blotting (Fig 3).

Define what it is the percentage of splicing inclusion (PSI).

PSI (percent spliced in) is the percentage of transcripts that include a specific exon. In this case the percentage of SS4+ of the total RNA splice variants (SS4- + SS4+). It's a measure of how an exon is included in the final mRNA. This point has been added in the text.

At 2.5. Western Blotting section, specify which hippocampal tissues were analyzed with the technique. Were they protein extracts from hippocampal neurons, hippocampal slices, hippocampus?  Indicate it.

For western analysis, protein extracts were obtained from hippocampal slices as reported in Fig 1B or from hippocampal neurons isolated from C57BL/6 male mice at E17.5 age, as reported in Fig 3A. These informations have been added in the text and reported in the relative figure legends.

At 2.6. section specify which hippocampal tissue was analyzed with the CB1 immunohistochemistry technique. The concept of using sections from the hippocampus is not without its ambiguities. In figure 1 authors indicated: A) Parasagittal hippocampal section from one month old WT mice. What tissue was used in this technique? Why sections were then counterstained with hematoxylin? Indicate it.

We thank the reviewer for this comment. We noticed the lack of clarity in the text and we modified the text. We clarified that in immunohistochemistry experiments we used whole brain from one month old mice. Brains were perfused with 4% PFA and paraffin embedded. Section of 5 mm were cut sagittally. Sections were counterstained with hematoxylin to identify the nuclei in the sections. The figure 1A has been modified

Figure 1. A) Enhance the image. Indicate with arrows which cells express CB1 receptors. Include a microphotograph of the entire slice. B) What does the first column mean? Brain? What does that band correspond to? Authors mention: Tubulin is used as reference gene. As reference gene? Or protein?  Indicate it.

The Figure 1A has been replaced with a new figure in which a microphotograph of the entire section of the brain has been added. White arrowheads are added to indicate cells expressing CB1 receptors.

In Figure 1B the first column reports the CB1 band detected in whole brain extract that is used as positive control. Tubulin is used as reference of protein loading.

Legend of Figure 1 has been modified accordingly.

At figure 3: A) Time course analysis of Akt phosphorylation in hippocampal neurons treated with ACEA (5’, 15’, 30’) or with AM251 alone and in combination with ACEA (40’ before agonist). However, at section 2.1. Primary hippocampal neuronal cultures authors mention that hippocampal neuronal cultures were treated with vehicle (DMSO) and drugs ACEA (1μM) or AM251(2 μM) for 16 hours. At what time were the hippocampal neuron cultures actually treated? 16 hours or 5’, 15’, 30’ and 40’ before agonist? Explain and indicate it correctly. Also at Figure 3 section B is missing. Authors mention “Tubulin is used as normalizer”, What it means normalizer? Correct it.

We are sorry for this mistake. As we report previosly, we have never performed a 16-hour treatment in our experiments on neuronal cultures. Figure 3 A shows the experiment performed on cultured hippocampal neurons to study the activated intracellular pathway. The relative legend reports the time course analysis of Akt phosphorylation in hippocampal neurons treated with ACEA, (5’, 15’, 30’) or with AM251 alone and in combination with ACEA (40’ before agonist). In panel B is reported the histogram showing the densitometric analysis of western blotting. We introduced these modifications in the text and in the figure legend.

Why were different drug administration times used (1 or 6 hours, 5-15-30-40 minutes, 16 hours? Justify your answer.

The different drug administration times used in the study—ranging from minutes (5–15–30–40 min) to hours (1 h, 6 h, 16 h)—were likely selected to capture the temporal dynamics of distinct biological processes affected by CB1 receptor modulation. Here’s a justified rationale for each timeframe based on the type of assay and molecular mechanism involved:

  1. Short-Term Treatments (5, 15, 30, 40 minutes)

These short time points were used primarily for Western blot analysis of intracellular signaling, particularly Akt phosphorylation (Fig. 3). Such signaling events are rapid and transient, typically occurring within minutes after receptor activation. Thus, the chosen time course allows monitoring the onset and duration of CB1-triggered PI3K/Akt pathway activation in hippocampal neurons. A shorter treatment time was used to better preserve the neural vitality of the slices during the recording process. This shorter duration (1 hour) was considered sufficient to trigger and observe the acute drug effect on synaptic transmission

  1. Intermediate-Term Treatments (1 hour)

The 1-hour treatment was applied before electrophysiological recordings (e.g., evoked and spontaneous EPSCs), as seen in Figures 2 and 6. This duration ensures sufficient time for drug diffusion and receptor interaction, while still reflecting acute synaptic responses without inducing compensatory homeostatic changes that might occur with prolonged exposure.

  1. Long-Term Treatments (6 hours)

The 6-hour treatment was chosen for molecular assays such as RT-PCR, RT-qPCR, and alternative splicing analysis of neurexins (e.g., Fig. 4 and 5). Changes in gene expression and splicing patterns require longer timeframes to allow transcriptional and post-transcriptional mechanisms to occur. This window is ideal to observe CB1-mediated modulation of alternative splicing via the SLM2 splicing factor.

We apologise for error in the text reporting 16-hour treatment. This timing was not used for experiments.

Figure 4. Correct in B) NRXS 2 and 3 graphs, the numbers on the y axis.

The numbers on the y axis have been corrected.

Line 441-443. Please include the figure 7. It was inadvertently omitted in the writing.

Figure 7 is included at line 462-464.

Authors do not discuss anything regarding the Akt phosphorylation in hippocampal neurons. Include the information in the discussion section.

This point has been reported in the discussion (Lines 423-428): “First of all, our data indicate the presence of a functionally active CB1 receptor in hippo-campus. We demonstrated that pharmacological stimulation with agonist/antagonist of CB1 in hippocampal slices modulated neurotransmission. In addition, we found that treatment of isolated hippocampal neurons with a CB1-selective agonist ACEA activated the PI3K pathway by increasing Akt phosphorylation in a way attenuated by a CB1-selective antagonist AM251”.

Additionally, authors should discuss the limitations of their study, which were determined using ex vivo hippocampal slices. It is important to note that the results obtained in this study could vary when using an in vivo model with an intact animal.

Following reviewer suggestion the following statement have been added to discussion

While our findings provide novel insights into the modulation of neurexin alternative splicing by CB1 receptor signaling, some limitations must be acknowledged. Most notably, the experiments were conducted using ex vivo hippocampal slices, which, although preserving some native connectivity and architecture, do not fully recapitulate the complexity of the intact brain. In vivo systems involve intricate interactions among multiple brain regions, neuromodulatory inputs, vascular dynamics, and behavioral states that cannot be entirely reproduced ex vivo. Therefore, it is possible that the extent and dynamics of CB1-dependent modulation of neurexin splicing—and its downstream effects on synaptic transmission—could differ in an intact organism. Further in vivo studies will be necessary to validate whether the observed mechanisms contribute similarly to neuronal function and synaptic plasticity under physiological or pathological conditions.

Reviewer 2 Report

Comments and Suggestions for Authors

The manuscript titled “Modulation of Neurexins Alternative Splicing by Cannabinoid 2 Receptors Signaling” presents compelling evidence for a novel mechanism linking cannabinoid CB1 receptor activity to alternative splicing of neurexin genes in the hippocampus. The authors employ ex vivo hippocampal slices and pharmacological agents to demonstrate that activation or inhibition of CB1 modulates the inclusion of the SS4 exon in Nrxn1-3, thereby influencing synaptic transmission.

The work is well-designed and methodologically sound, integrating electrophysiological recordings, RT-PCR, and genetic knockout models. The findings are significant, suggesting that the endocannabinoid system dynamically regulates synaptic plasticity via splicing factors like SLM2. Notably, SLM2 knockout abolished drug-induced splicing changes and neurotransmission modulation, strengthening the proposed mechanistic link.

Overall, this study contributes important insights into the intersection of neuromodulation, gene expression, and memory-related plasticity, offering potential implications for understanding disorders such as depression and autism.

My comments:

  1. While the study convincingly shows that CB1 signaling modulates SS4 alternative splicing in Nrxn1-3, it does not provide direct functional validation of how the different splice isoforms (SS4+ vs SS4−) affect synaptic transmission at the molecular or behavioral level. The authors cite literature suggesting their roles, but a more targeted approach (e.g., isoform-specific overexpression or knockdown) would significantly strengthen the causal link.
  2. figure no. 1 is of poor quality. There is no information from the hippocampal cortex, does this fragment come from the dorsal or ventral side? I think it would be advisable to mark the remaining hippocampal fields on the photo.
    methodology: were the preparations heat -treated to expose the antigen? what was the thickness of the sections? how many mice were used in the experiment?

Author Response

Response to Reviewers

We would like to thank the reviewers for the suggestions provided to improve the work, below are our responses to the points raised by the reviewers

REF 2

Comments and Suggestions for Authors

The manuscript titled “Modulation of Neurexins Alternative Splicing by Cannabinoid 2 Receptors Signaling” presents compelling evidence for a novel mechanism linking cannabinoid CB1 receptor activity to alternative splicing of neurexin genes in the hippocampus. The authors employ ex vivo hippocampal slices and pharmacological agents to demonstrate that activation or inhibition of CB1 modulates the inclusion of the SS4 exon in Nrxn1-3, thereby influencing synaptic transmission.

The work is well-designed and methodologically sound, integrating electrophysiological recordings, RT-PCR, and genetic knockout models. The findings are significant, suggesting that the endocannabinoid system dynamically regulates synaptic plasticity via splicing factors like SLM2. Notably, SLM2 knockout abolished drug-induced splicing changes and neurotransmission modulation, strengthening the proposed mechanistic link.

Overall, this study contributes important insights into the intersection of neuromodulation, gene expression, and memory-related plasticity, offering potential implications for understanding disorders such as depression and autism.

My comments:

  1. While the study convincingly shows that CB1 signaling modulates SS4 alternative splicing in Nrxn1-3, it does not provide direct functional validation of how the different splice isoforms (SS4+ vs SS4−) affect synaptic transmission at the molecular or behavioral level. The authors cite literature suggesting their roles, but a more targeted approach (e.g., isoform-specific overexpression or knockdown) would significantly strengthen the causal link.

We agree with the referee. This is an important point that could shed light on the functional role of the two splice variants. Future studies will be focused to investigate this link.

  1. figure no. 1 is of poor quality. There is no information from the hippocampal cortex, does this fragment come from the dorsal or ventral side? I think it would be advisable to mark the remaining hippocampal fields on the photo.
    methodology: were the preparations heat -treated to expose the antigen? what was the thickness of the sections? how many mice were used in the experiment?

We thank the reviewer for this comment. We have now replaced Figure 1A and we reported the details in the figure legend and in the 2.7 section of materials and methods. The whole brain from one month old mice were obtained after perfusion with 4% PFA and paraffin embedded. Section of 5mm were cut sagittally. The preparations were heat-treated to expose the antigen. Sections were counterstained with hematoxylin to identify the nuclei in the sections. The figure shows a parasagittal image of ventral hippocampus. Brains are from n=3 mice.

Round 2

Reviewer 1 Report

Comments and Suggestions for Authors

What do the calibration bars in Figure 1a correspond to? Indicate it.

FIGURE 7 IS STILL MISSING IN THE MANUSCRIPT. INCLUDE IT.

Include these information in the methods section:

The different drug administration times used in the study—ranging from minutes (5–15–30–40 min) to hours (1 h, 6 h, 16 h)—were likely selected to capture the temporal dynamics of distinct biological processes affected by CB1 receptor modulation. Here’s a justified rationale for each timeframe based on the type of assay and molecular mechanism involved:

Short-Term Treatments (5, 15, 30, 40 minutes)

These short time points were used primarily for Western blot analysis of intracellular signaling, particularly Akt phosphorylation (Fig. 3). Such signaling events are rapid and transient, typically occurring within minutes after receptor activation. Thus, the chosen time course allows monitoring the onset and duration of CB1-triggered PI3K/Akt pathway activation in hippocampal neurons. A shorter treatment time was used to better preserve the neural vitality of the slices during the recording process. This shorter duration (1 hour) was considered sufficient to trigger and observe the acute drug effect on synaptic transmission

Intermediate-Term Treatments (1 hour)

The 1-hour treatment was applied before electrophysiological recordings (e.g., evoked and spontaneous EPSCs), as seen in Figures 2 and 6. This duration ensures sufficient time for drug diffusion and receptor interaction, while still reflecting acute synaptic responses without inducing compensatory homeostatic changes that might occur with prolonged exposure.

Long-Term Treatments (6 hours)

The 6-hour treatment was chosen for molecular assays such as RT-PCR, RT-qPCR, and alternative splicing analysis of neurexins (e.g., Fig. 4 and 5). Changes in gene expression and splicing patterns require longer timeframes to allow transcriptional and post-transcriptional mechanisms to occur. This window is ideal to observe CB1-mediated modulation of alternative splicing via the SLM2 splicing factor.

Author Response

Point-by-point response to the reviewer’s comments:

Comment 1: What do the calibration bars in Figure 1a correspond to? Indicate it.

Response 1: Thank you for pointing this out. The scale bar value has been included in the legend of Figure 1

Comment 2: FIGURE 7 IS STILL MISSING IN THE MANUSCRIPT. INCLUDE IT.

Response 2: Thank you for the comment. Figure 7 shows the proposed model on the basis of our results. Figure 7 needs to be included in the Discussion section. The exact point of insertion has been clearly indicated in the text at line 488. We uploaded again this figure 7.

Comment 3: Include these information in the methods section:

Response 3: Thank you for pointing this out. All relevant details have been appropriately incorporated into the manuscript in Materials and Methods .

In addition:

The text has been carefully revised.

As requested Figure 5A has been changed to ensure consistency between the original image and the figure in the manuscript. The new figure 5 has been uploaded again. The original has been sent previously by e-mail.